

# Optimized nitrogen fertilizer application strategies under supplementary irrigation improved winter wheat (*Triticum aestivum L.*) yield and grain protein yield

Zhen Zhang, Zhenwen Yu, Yongli Zhang and Yu Shi

Shandong Agricultural University, Taian, China

## ABSTRACT

**Background**. Exploring suitable split nitrogen management is essential for winter wheat production in the Huang-Huai-Hai Plain of China (HPC) under water-saving irrigation conditions, which can increase grain and protein yields by improving nitrogen translocation, metabolic enzyme activity and grain nitrogen accumulation.

**Methods**. Therefore, a 2-year field experiment was conducted to investigate these effects in HPC. Nitrogen fertilizer was applied at a constant total rate (240 kg/ha), split between the sowing and at winter wheat jointing growth stage in varying ratios, N1 (0% basal and 100% dressing fertilizer), N2 (30% basal and 70% dressing fertilizer), N3 (50% basal and 50% dressing fertilizer), N4 (70% basal and 30% dressing fertilizer), and N5 (100% basal and 0% dressing fertilizer).

**Results**. We found that the N3 treatment significantly increased nitrogen accumulation post-anthesis and nitrogen translocation to grains. In addition, this treatment significantly increased flag leaf free amino acid levels, and nitrate reductase and glutamine synthetase activities, as well as the accumulation rate, active accumulation period, and accumulation of 1000-grain nitrogen. These factors all contributed to high grain nitrogen accumulation. Finally, grain yield increase due to N3 ranging from 5.3% to 15.4% and protein yield from 13.7% to 31.6%. The grain and protein yields were significantly and positively correlated with nitrogen transport parameters, nitrogen metabolic enzyme activity levels, grain nitrogen filling parameters.

**Conclusions**. Therefore, the use of split nitrogen fertilizer application at a ratio of 50%:50% basal-topdressing is recommended for supporting high grain protein levels and strong nitrogen translocation, in pursuit of high-quality grain yield.

# INTRODUCTION

Chinese agriculture currently faces two major challenges: the excessive application of nitrogen fertilizer and increasingly widespread water shortages (*Wu et al., 2020*). Wheat is a crucial and global food source that supplies over 45% of calories and over 40% of protein available to the global population. In China, wheat is mostly grown under traditional flood irrigation, which produces high grain yields but requires large quantities of freshwater (*Li et*

Corresponding author
Yongli Zhang,
zhangylsdau@sohu.com,
zhangyl@sdau.edu.cn

*al., 2020*). Specifically, wheat irrigation accounts for approximately 50% of all agricultural water use in China (*Chen et al., 2020*). Therefore, water-saving irrigation technologies are essential for the development of sustainable wheat production system. *Man et al. (2014)* for example, reported a water-saving cultivation technique for supplementary irrigation based on soil moisture measurements at critical stages, which, compared with traditional quantitative irrigation, produced a better crop growth and yield advantage with a lower irrigation input.

Nitrogen is one of the most important nutrients for wheat production (*Zhou et al., 2018*). Nitrogen fertilizer application is generally the most effective way to increase plant nitrogen accumulation and grain yield, and to improve the grain protein content and other quality indicators (*Ma et al., 2019*). However, farmers often misinterpret the relationship between grain yield and nitrogen fertilization, and thus overestimate the yield benefits. This results in excessive nitrogen fertilizer application in agricultural production (*Wu & Ma, 2015*; *Gao et al., 2020*). In fact, excessive nitrogen application does not significantly improve the grain yield once the levels have exceeded the ability of plants to uptake nitrate, and instead the residual nitrogen fertilizer in the soil is lost via nitrate leaching and nitrous oxide emissions (*Wang et al., 2020*). Therefore. it is necessary to improve nitrogen fertilization strategies without increasing the fertilizer application.

*Rajinder et al. (2017)* showed that split applications of nitrogen fertilizer may help to meet the nitrogen demand via soil supply and to enhance crop yield, grain quality, and fertilizer use efficiency. For this reason, appropriate split nitrogen applications are the most critical component of nitrogen fertilizer management (*Kandel, Gowda & Northup, 2020*; *Li et al., 2021*). Previous studies have shown that pre-jointing nitrogen accumulation increased with an increase in the basal nitrogen rate, while post-jointing nitrogen accumulation increased with the increase in the topdressing nitrogen rate at a total fertilizer application rate of 210 kg/ha (*Shi et al., 2012*). Similarly, *Tian et al. (2018)*. found that an increase in the topdressing nitrogen fertilizer ratio could improve the accumulation of nitrogen after the jointing stage. *Xu et al. (2018)* found that the nitrogen accumulation in above-ground plant parts and grain at maturity was higher under a 4:6 of basal-topdressing ratio treatment of nitrogen fertilizer by an average of 35.76%, 24.13% and 37.36%, 19.27% compared with the 6:4 and 5:5 basal-topdressing ratio with a total nitrogen application rate of 210 kg/ha. However, *Shi et al. (2012)*. reported that plant nitrogen uptake between sowing and maturity was significantly higher with a 5:5 basal-topdressing regime than a 3:7 regime; however, this effect was not significantly different with basal-topdressing ratios of 5:5 and 7:3. Although many studies have investigated split nitrogen management, no consensus has been reached regarding the optimum basal-topdressing ratio of nitrogen fertilizer. However, there are only a few studies on grain nitrogen filling characteristics and nitrogen metabolizing enzyme activity which are both important to plant nitrogen accumulation. Furthermore, only a studies have investigated the effect of split nitrogen application under supplemental irrigation that is based on soil moisture measurements. In addition, remobilized nitrogen from vegetative organs and its accumulation after anthesis are both potential sources of for grain components (*Nehe et al., 2018a*). Therefore, to

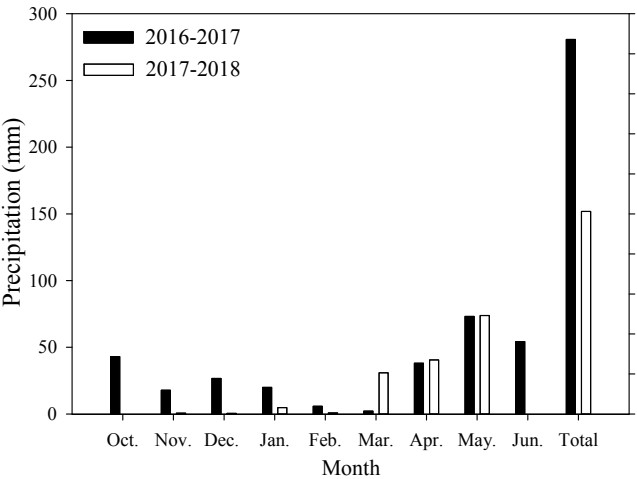

**Figure 1  Monthly precipitation during wheat growth period.** Each data point represents the total monthly rainfall.

effectively manage fertilizer applications, it is important to quantify how these sources contribute to the nitrogen content of wheat grain.

The objectives of this study were to (1) evaluate the effect of split nitrogen fertilizer applications on the grain and grain protein yields, (2) identify how split nitrogen fertilizer affects the translocation of nitrogen from vegetative organs to the grain and the nitrogen accumulation in the grain after anthesis, and (3) determine the relationship between grain protein and nitrogen translocation from the vegetative organs to the grain.

## MATERIALS & METHODS

### Location and timing

Field experiments were conducted at Xiaomeng Town, Jining City, Shandong Province, in 2016 and 2017. This experimental area is located in the center of the Huang-Huai-Hai Plain, which has an environment that is typical and representative of the plain. The area has a temperate continental climate, an annual average temperature of 13.6 °C, an annual accumulated sunshine period of 2460.9 h, an annual precipitation range from 480 to 850 mm, and an annual average precipitation of 621.2 mm. The soil in the region is classified as loam. Prior to the experiment, the soil organic matter, total nitrogen, available nitrogen, potassium and phosphorus contents were 14.2 g/kg, 1.1 g/kg, 122.6 mg/kg, 129.4 mg/kg, and 38.1 mg/kg, respectively, at the 0∼20 cm soil depth. The monthly average precipitation data of the study area are shown in Fig. 1.

### Experimental design, material, and growing conditions

Experiments were conducted in a randomized complete block design with three replicates per treatment. Each treatment plot was 20 m$^2$. Adjacent plots of different nitrogen application treatments were separated by a 1.5 m gap to avoid interference between treatments. We used five treatments with different basal-topdressing fertilizer ratios (i.e.,

0% basal and 100% dressing fertilizer, 30% basal and 70% dressing fertilizer, 50% basal and 50% dressing fertilizer, 70% basal and 30% dressing fertilizer, 100% basal and 0% dressing fertilizer, hereafter referred to as N1, N2, N3, N4 and N5, respectively), with a nitrogen application rate of 240 kg/ha. Soil moisture management in the experimental plot was based on the water-saving cultivation techniques for supplementary irrigation by measuring the soil moisture. The relative water content in the 0~40cm soil layer was measured and then supplemental to 70% at the wheat jointing and flowering stages. The irrigation was calculated according to the formula: M = 10×r ×H ×($\beta$i- $\beta$j). H stands for the pseudo-moist soil depth. r represents the bulk density of the quasi-moist soil depth. $\beta$i stands for desired water content. And the $\beta$j is the water content of the soil before irrigation. A hose was used for all irrigation and to measure irrigation in meters (*Man et al., 2014*).

In this study, we assessed the most widely used winter wheat variety in the surrounding area (cv. Jimai 22). Jimai 22 was bred at the Shandong Academy of Agricultural Sciences. It was the first variety planted in China for nine years in a row, with a cumulative planting area of 270 million mu and an average growth period of 231.4 days. Urea (N 46%), calcium super phosphate ($P_2O_5$ 12%) and potassium chloride ($K_2O$ 60%) were the sources of nitrogen, phosphorus and potassium, respectively. The amount of phosphate fertilizer and potash fertilizer were 150 kg/ha and 112.5 kg/ha respectively. Before winter wheat was sown, the basal nitrogen fertilizer and all of the phosphate and potash fertilizers were spread onto the soil surface and immediately mixed with the soil to a depth of 20 cm using a rotary cultivator. After fertilizer application, the seeds were sown on October 12, 2016 and October 24, 2017, at a depth of 3~5cm, and at a rate of 1.8 million per square hectometer. During the jointing stage, furrows were created for nitrogen fertilizer application and then immediately covered (179 days after sowing in 2016-2017 and 170 days after sowing in 2017–2018). Grain was harvested on June 8, 2017 and June 7, 2018. Weeds, diseases, and pests were controlled adequately during both growing seasons to avoid the influence of any other growth limiting factors.

## Sampling and analysis
### Nitrogen accumulation and transport
Twenty representative plant samples were randomly collected from each plot at the anthesis and maturity stages. Samples were separated into leaves, stems + sheaths and spikes at the anthesis stage, they were separated into leaves, stems+sheaths, spike axes + grain husks, and grain at the maturity stage. These samples were oven-dried at 105 °C for 30 min and then at 80 °C to constant mass before they were weighed to determine the biomass. All wheat samples were crushed using a ball mill before nutrient analysis. Nitrogen concentrations were determined following the method of *Liu et al. (2019)*. Nitrogen accumulation amount (NAA) was calculated as the product of dry matter accumulation and nitrogen concentration. Nitrogen translocation parameters were calculated from the accumulated nitrogen in the vegetative organs at anthesis and maturity. Four indicators were used to evaluate the nitrogen translocation (*Liu et al., 2019*): (1) nitrogen translocation amount (NTA), (2) nitrogen translocation efficiency (TE), (3) contribution proportion

(CP) and (4) nitrogen accumulation amount after anthesis (NAFA). They were calculated using the following formulas:

$$NTA = NAAA - NAAM \qquad (1)$$

$$TE = NTA/NAAA \qquad (2)$$

$$CP = NTA/GNAA \qquad (3)$$

$$NAFA = (GNAA - NTA)/GNAA \qquad (4)$$

where ANAA is the nitrogen accumulation in the vegetative organs at anthesis, MNAA is the nitrogen accumulation in the vegetative organs at maturity, and GNAA is the nitrogen accumulation in the grain at maturity.

### N-metabolizing enzymes

The flag leaves of 20 randomly selected wheat plants were collected at 8:00-9:00 A.M. between 0 to 35 days after anthesis (DAA), which provided six sampling times altogether in both growing seasons of each year (2016 and 2017), respectively. The sampled flag leaves were packed into plastic bags, transported to the laboratory in liquid nitrogen, and then stored at $-80\,°C$ for analysis (*Oosten et al., 2019*). Half of the sampled flag leaves were used to measure the nitrate reductase (NR) activity, while the remaining were used to measure the glutamine synthetase (GS) activity.

The NR activities were assayed using the protocols described by *Hu et al. (2016)*. Briefly, the NR in flag leaf samples (0.3 g) was extracted in 4 ml extraction buffer 0.1 phosphate buffer ($NaH_2PO_4 \cdot 2H_2O$ and $Na_2HPO_4 \cdot 12H_2O$, pH of 7.5). The extracted samples were centrifuged at 12,000 rpm for 20 min at 4 °C. Then, 0.4 ml extraction solution was mixed with 0.4 ml nicotinamide adenine dinucleotide (NADH) and 1.2 ml 0.1 mol/L $KNO_3$. The mixture was allowed to react at 25 °C for 30 min. For the control solution, NADH was replaced with 0.4 ml 0.1 mol/L sodium phosphate. The reaction was terminated with the addition of 1 ml sulphanilamide; after which, 1 ml 1% N-1-naphthylethylenediamine dihydrochloride was added to the mixture. The color of the mixture was allowed to develop for 15 min. The mixture was then centrifuged at 12,000 rpm for 10 min and its absorbance measured at 540 nm using a spectrophotometer. The NR activity was calculated from the nitrite nitrogen standard curve.

Additional methods were used to assess the activity of GS (*Hu et al., 2016*). Briefly, GS in flag leaf samples (0.2 g) was extracted in 3 ml extraction buffer (50 mmol/L sodium phosphate buffer, had a pH of 7.2, 50 mmol/L $Na_2SO_4$ and 0.5 mmol $L^{-1}$ $Na_2EDTA$). The mixture was centrifuged at 20,000 rpm for 20 min at 4 °C. We mixed 1.2 ml supernatant with 0.3 ml 0.3 mol/L Na-Glu, 0.6 ml 0.25 mol/L imidazole-HCl (pH 7.0), 0.2 ml 0.5 mol/L $MgSO_4$ and 0.4 ml 0.03 mol/L Na-adenosine triphosphate (ATP) (pH 7.0). This mixture was incubated at 25 °C for 5 min before 0.2 ml 1.0 mol/L hydroxylamine was added, followed by incubation for 20 min. The reaction was terminated using 0.8 ml mixed reagent (10% $FeCl_3 \cdot 6H_2O$, 50% (v/v) HCl and 24% (w/v) trichloroacetic acid). The color of the mixture was allowed to develop for 20 min. The mixture was then centrifuged at 15,000 rpm for 10 min at 25 °C before its absorbance was measured at 540 nm. The GS activity was calculated from the standard curve of c-glutamyl-hydroxamate.

### Amino acids measurement

Twenty fresh flag leaf and grain samples were collected every 7 days during the periods between anthesis to maturity (sampling 6 times altogether, every year from 2016 to 2018). The free amino acids in these samples were determined according to the method described by *Ali et al. (2019)*. All fresh leaf tissue was thoroughly mixed with phosphate buffer and the solutions were filtered to produce extracts for analysis. For each sample, 1 ml 10% pyridine and 1 ml 2% sodium hydrochloride solutions were added to 1 ml extract in a 25 ml tube and heated in boiling water for 30 min. The volume of each test tube was made up to 50 ml with distilled water. Sample absorption was measured at 570 nm. The amino acid content was calculated using a standard curve for l-leucine.

### Nitrogen accumulation and fitting equation in 1,000-grain

According to the method described by *Zhang et al. (2011)*, we selected wheat stems that reached the anthesis stage on the same date and marked these as single stems with similar growth. In each plot, 200 single stems were marked to ensure uniformity. 20 randomly labeled spikes were collected between 09:00 and 11:00 A.M. in 7-day intervals (sampling five times altogether, every year from 2016 to 2018) from 7 days after anthesis (DAA) until harvest. All harvested spikes were placed in marked paper bags and dried in an oven at 105 °C for 30 min and then at 75 °C to a constant weight. Thereafter, the grain was gently threshed from the ears by hand and the material was carefully divided into the grains and the other parts of the ear. Grain samples were weighed to determine the 1000-grain weight, and then crushed using a ball mill, as described previously, for nutrient analysis. The nitrogen concentrations of the samples were determined using the method described by Zhang. The 1000-grain accumulated nitrogen content was calculated from the 1000-grain weight and nitrogen concentration. Previous studies have suggested that Logistic growth equation can be fitted with 1000-grain weight after anthesis to evaluate the effects of treatments on the grain-filling process. In the present study, we used this method to evaluate the treatment effects on the nitrogen accumulation of grains. The logistic growth equation was used to fit the process of grain nitrogen accumulation, and the base parameters were measured as follows *Zhang et al. (2011)*:

$$y = a/(1 + be^{-cx})$$

Where $y$ is the nitrogen accumulation (mg), $a$ is the final nitrogen accumulation (mg), x is days after flowering, and b and c are the parameters set by the regression equation. In this analysis, we adopted the following secondary parameters to describe the filling characteristics: Vmax (the maximum accumulation rate), Vmean (the average accumulation rate) and D (the active accumulation period).

### The content of grain protein and grain protein yield at maturity

Grain samples were collected at maturity. The protein content of the samples was determined using the Anna-Catuarina method (*Anna-Catuarina, Britta & Karl, 2018*), whereby the total protein content of the grain was equal to its nitrogen content multiplied by 5.7. This coefficient was used to calculate the grain protein yield from the measured grain yield and its total protein content.

*Grain yield*

After the grains reached complete maturity, which was marked by the appearance of grain (*Wei et al., 2017*), we harvested the spikes from 2 m² areas in each plot and then determined the grain weight. Specifically, the spikes were threshed using a small wheat thresher (Shandong zhongfa Mechanical Equipment Co. Ltd, China), and the grains were air-dried and weighed. Grain yield per hectare was estimated according to the grain weight of the total kernels from the 2 m² sampled areas.

## Statistical analysis

All data were analyzed using two-way analysis of variance (ANOVA) in the SPSS 13.0 statistical software package to test whether significant differences existed between the split nitrogen treatments in both growth seasons.The means for the treatment was compared using the least significant difference (LSD) test at a probability level ≤ 0.05. The SPSS 13.0 statistical program was also used to fit the equation for nitrogen accumulation in grains to obtain the parameters Vmax, Vmean and D. Pearson's correlation analysis was used to identify possible relationships between grain yield, grain protein yield and other parameters. All charts were plotted using SigmaPlot 12.5.

## RESULTS

### Nitrogen accumulation amount in vegetative organs at anthesis and maturity stages

Split nitrogen treatments had significant effects on the nitrogen accumulation in the leaf, stem + sheath, and spike axis + grain husk tissues at the anthesis and maturity stages (Table 1). In the first growing season, the nitrogen accumulation amount in leaves of the N3 treatment was significantly higher than those in the N1, N4 and N5 treatment, while there was no significant difference between the N3 and N2 treatments. The nitrogen accumulation in stem and sheath tissues of the N3 treatment was significantly higher than that of N5 treatment, and there was no significant difference in N1, N2, N3 and N4 treatment. The nitrogen accumulation amount in spike axis and grain husk were the highest under the N1 and N2 treatments, followed by N3 and N4, and finally N5 treatment. The nitrogen accumulation amount in vegetative organs at maturity increased with the amount of topdressing nitrogen. The N3 treatment substantially increased nitrogen accumulation in vegetative organs at anthesis, especially in leaf and stem + sheath tissues.

### Nitrogen translocation from vegetative organs to grain

The translocation of nitrogen from vegetative organs to grain after anthesis varied among the split nitrogen treatments (Table 2). In the first growing season, the amount of nitrogen thst translocated was the highest in the N3 treatment, followed by the N2 and N4, and then the N1 and N5 treatments. The nitrogen accumulation after anthesis with N3 treatment was significantly higher than that measured in the N4 and N5 treatments, while there was no significant difference in N1, N2 and N3 treatments. More nitrogen was translocated from the leaves and stems + sheaths (which in turn contributed more to the grain nitrogen) than the reproductive organs, i.e., the spike axis + grain husks. This indicated that the N3

**Table 1  Effects of different treatments on nitrogen accumulation amount in organs at anthesis and maturity stages.**

| Treatments | | Nitrogen accumulation amount in leaf kg/ha | | Nitrogen accumulation amount in stem and sheath kg/ha | | Nitrogen accumulation amount in spike axis and grain husk kg/ha | |
|---|---|---|---|---|---|---|---|
| | | Anthesis | Maturity | Anthesis | Maturity | Anthesis | Maturity |
| 2016 | N1 | 82.5b | 18.5a | 103.7a | 44.2a | 29.7a | 15.4a |
| | N2 | 88.3a | 17.1b | 105.6a | 39.5b | 29.9a | 15.2a |
| | N3 | 91.1a | 14.5c | 106.5a | 35.4c | 27.2b | 11.9b |
| | N4 | 82.3b | 13.8d | 100.5a | 33.5d | 27.6b | 11.9b |
| | N5 | 70.1c | 12.7e | 93.0b | 33.7d | 24.9c | 10.8c |
| 2017 | N1 | 84.2b | 21.5a | 95.2a | 45.2a | 30.6a | 20.8a |
| | N2 | 91.2a | 19.0b | 98.2a | 40.4b | 28.8b | 18.6b |
| | N3 | 90.8a | 16.1c | 95.3a | 32.4c | 25.7c | 14.7c |
| | N4 | 83.6b | 15.6c | 89.4b | 31.0c | 26.2c | 14.3c |
| | N5 | 73.2c | 14.5d | 82.7c | 29.9c | 25.7c | 13.8c |

**Notes.**
*, ** significant at the 0.05 and 0.01 probability levels, respectively, ns, no significant. Each data point represents the average of the measured data.

treatment significantly increased the translocation of nitrogen from vegetative organs to the grain after anthesis, especially from the leaf and stem + sheath tissues, and that the nitrogen accumulation increased after anthesis. This was probably the key reason why the N3 treatment results in the higher accumulation of grain nitrogen.

## Enzyme activity levels of flag leaf after anthesis

Changes in NR and GS activity levels in the flag leaves after anthesis are shown in Fig. 2. The ANOVA results showed that the split nitrogen treatments had significant effects on the flag leaf NR and GS activity levels. Flag leaf NR activity levels were not significantly different across treatments from 0 to 7 DAA. However, from 14 to 35 DAA, the N3-treated plants showed the highest flag leaf NR activity, followed by N2- and N4- treated plants; the N1- and N5-treated plants showed the lowest NR activity. The results for GS activity were similar, suggesting that the N3 treatment improved the NR and GS activity levels in flag leaves.

## Free amino acid content of flag leaf after anthesis

Changes in the free amino acid contents in flag leaves and grain after anthesis are shown in Fig. 3. The ANOVA analysis showed that the split nitrogen treatments significantly affected the free amino acid contents. In the first growing season, the flag leaf free amino acid contents did not significantly differ between treatments at 0 DAA. The N3 treatment produced the highest levels from 7 to 35 DAA, followed by N2, N4, N1, N5, in decreasing order. Similarly, the free amino acid contents in the grain did not differ significantly between treatments at 7 DAA, but were significantly higher in the N3 treatment than in the unequal ratio treatments from 14 to 35 DAA, the latter of which did not differ from each other. Both growth seasons showed similar results, suggesting that the N3 treatment improved the nitrogen translocation in wheat after anthesis.

Zhang et al. (2021), *PeerJ*, DOI 10.7717/peerj.11467

**Table 2  Effects of different treatments on nitrogen translocation after anthesis from vegetative organs to grain.**

| Treatments | | Translation of nitrogen from vegetative organs to grain after anthesis | | | | | | | | | Nitrogen accumulation after anthesis | |
|---|---|---|---|---|---|---|---|---|---|---|---|---|
| | | Leaf | | | Stem and sheath | | | Spike axis and grain husk | | | NAFA kg/ha | CP % |
| | | NTA kg/ha | TE % | CP % | NTA kg/ha | TE % | CP % | NTA kg/ha | TE % | CP % | | |
| | N1 | 64.0c | 77.6b | 34.9a | 59.5c | 57.4b | 32.4a | 14.0b | 47.8b | 7.7b | 45.9a | 25.0a |
| | N2 | 71.2b | 80.7a | 35.6a | 66.1b | 62.6a | 33.0a | 14.8a | 49.3b | 7.4b | 48.0a | 24.0a |
| 2016 | N3 | 76.6a | 84.1a | 36.6a | 71.1a | 66.7a | 34.0a | 15.3a | 56.3a | 7.3b | 46.2a | 22.1b |
| | N4 | 68.6b | 83.3a | 35.4a | 67.0b | 66.6a | 34.6a | 15.7a | 56.9a | 8.1a | 42.4b | 21.9b |
| | N5 | 57.4d | 81.9a | 34.6a | 59.3c | 63.7a | 35.7a | 14.1b | 56.6a | 8.5a | 35.1c | 21.2b |
| | N1 | 62.7c | 74.4b | 37.7a | 50.1c | 52.6c | 30.1a | 9.8c | 31.9c | 5.9c | 43.8b | 26.3a |
| | N2 | 72.2a | 79.2a | 38.6a | 57.8b | 58.8b | 30.9a | 10.1c | 35.3c | 5.4c | 47.0a | 25.1a |
| 2017 | N3 | 74.8a | 82.3a | 38.2a | 62.9a | 66.0a | 32.2a | 11.0b | 42.7b | 5.6c | 46.9a | 24.0b |
| | N4 | 68.0b | 81.4a | 37.9a | 58.4b | 65.4a | 32.5a | 12.0a | 45.6a | 6.7b | 41.2c | 22.9b |
| | N5 | 58.8d | 80.3a | 37.2a | 52.8c | 63.9a | 33.5a | 11.9a | 46.3a | 7.5a | 34.4d | 21.8b |

Notes.

NTA, nitrogen translocation amount; TE, translocation efficiency; CP, contribution proportion; NAFA, nitrogen accumulation amount after anthesis.

*, ** significant at the 0.05 and 0.01 probability levels, respectively ns, no significant. Each data point represents the average of the measured data.

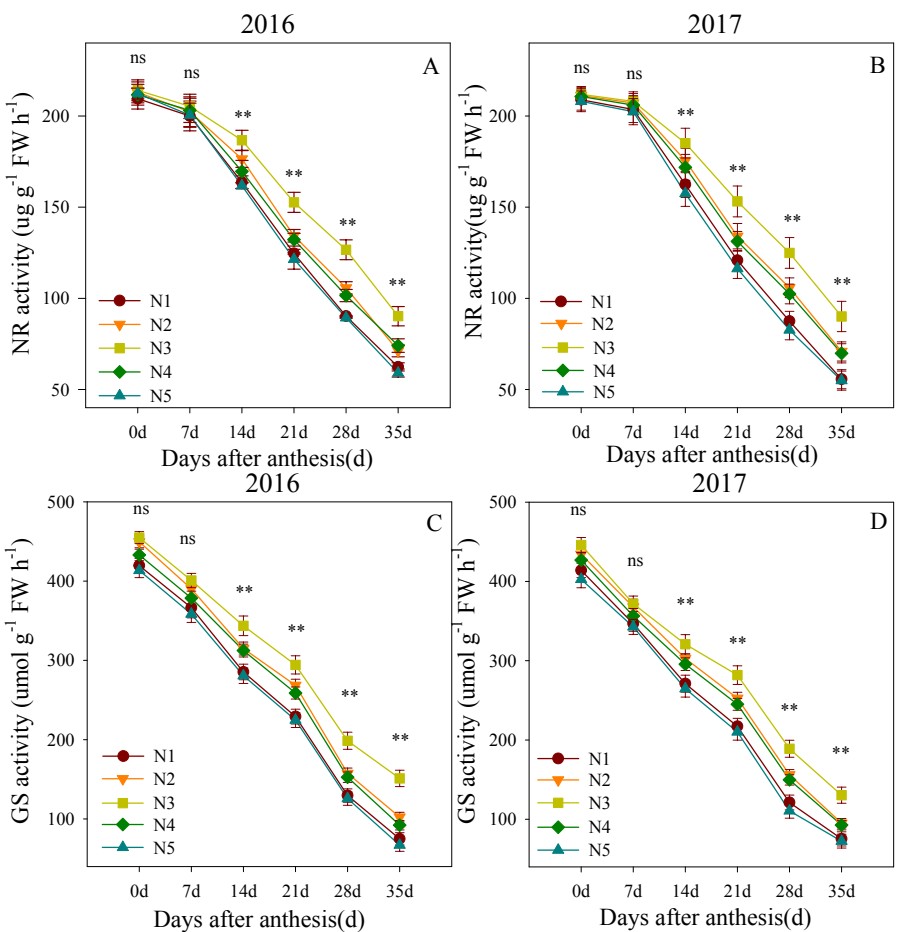

**Figure 2** (A–D) Effects of different treatments on NR activity and GS activity after anthesis. *, ** significant at the 0.05 and 0.01 probability levels, respectively, ns, no significant. Each data point represents the average of the measured data.

## Parameters of grain nitrogen filling stages after anthesis
### *Accumulation of 1000-grain nitrogen in grains*
The mitrogen accumulation in 1000-grain nitrogen after anthesis is shown in Fig. 4. In the first growing season, this outcome was not significantly different between treatments at 7 DAA. The N3 treatment produced the highest levels from 14 to 35 DAA, followed by N2, N4 and N1, with N5 producing the lowest levels. The two growing seasons presented similar results, suggesting that the N3 treatment improved the 1000-grain nitrogen accumulation in plants at the middle and late growing stages.

### *Grain nitrogen filling characteristics*
The fitted equations for grain nitrogen filling parameters after anthesis are shown in Table 3. The ANOVA results showed that split nitrogen treatments had significant effects on these equations. The maximum accumulation rate did not differ significantly between treatments in either growth season. In both years, the average accumulation rate and

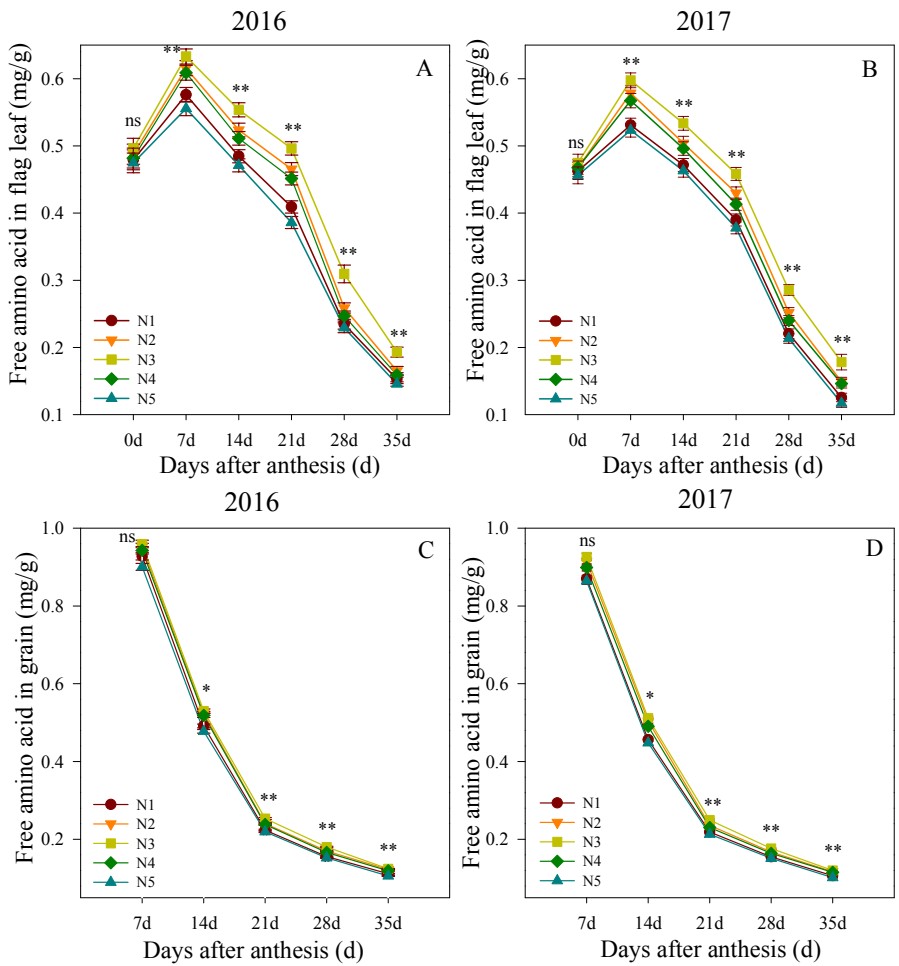

**Figure 3** (A–D) Effects of different treatments on free amino acid in flag leaf and grain after anthesis. *, ** significant at the 0.05 and 0.01 probability levels, respectively, ns, no significant. Each data point represents the average of the measured data.

active accumulation period in plants administered the N3 treatment were significantly higher than those administered unequal ratio treatments, between which there were no significant differences. This indicated that the N3 treatment significantly increased the average accumulation rate and active accumulation period after anthesis, providing another reason for why the N3 treatment resulted in the higher accumulation of grain nitrogen when compared with the other treatments tested.

## Grain yield and protein yield

The split nitrogen fertilizer treatments significantly affected the grain and protein yield in both growing seasons (Fig. 5). In the first growing season, the N3 treatment resulted in a grain yield that was on average 11.6%, 5.3%, 5.8% and 16.2% higher than that of the N1, N2, N4 and N5 treatments, respectively. Similarly, the N3 treatment resulted in a protein yield that was 14.0%, 10.0% and 26.0% higher than the N1, N4 and N5 treatments,

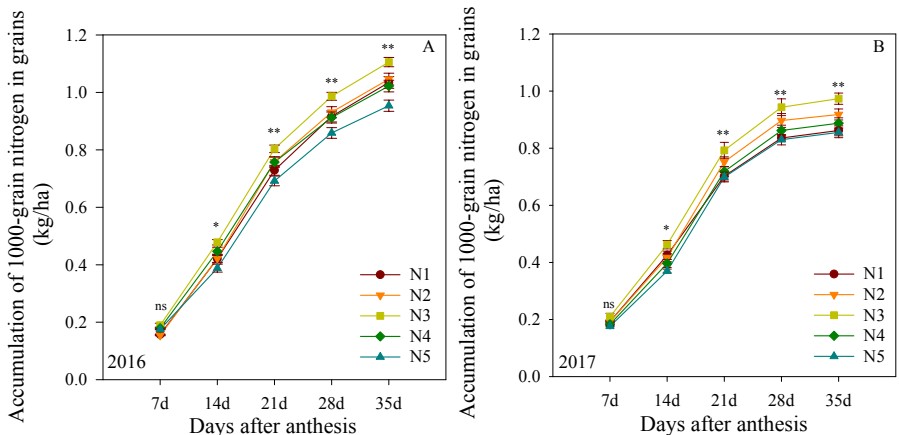

**Figure 4** (A–B) Effects of different treatments on accumulation of 1000-grain nitrogen in grains after anthesis. *, ** significant at the 0.05 and 0.01 probability levels, respectively, ns, no significant. Each data point represents the average of the measured data.

**Table 3** Effects of different treatments on nitrogen translocation after anthesis from vegetative organs to grain.

| Year | Treatments | Fitting equation of grain nitrogen filling | $R^2$ | Vmax | Vmean | D |
|------|-----------|--------------------------------------------|-------|------|-------|---|
| 2016 | N1 | $y = 1.066/(1 + 17.876e{-}0.179x)$ | 0.997 | 0.048a | 0.030b | 33.499b |
|  | N2 | $y = 1.065/(1 + 20.300e{-}0.178x)$ | 0.995 | 0.047a | 0.030b | 33.636b |
|  | N3 | $y = 1.139/(1 + 17.673e{-}0.171x)$ | 0.996 | 0.049a | 0.032a | 35.116a |
|  | N4 | $y = 1.042/(1 + 15.842e{-}0.179x)$ | 0.996 | 0.047a | 0.029b | 33.515b |
|  | N5 | $y = 0.991/(1 + 15.792e{-}0.185x)$ | 0.992 | 0.046a | 0.027c | 32.473b |
| 2017 | N1 | $y = 0.891/(1 + 14.568e{-}0.199x)$ | 0.994 | 0.044a | 0.025b | 30.189b |
|  | N2 | $y = 0.951/(1 + 18.322e{-}0.198x)$ | 0.995 | 0.047a | 0.026b | 30.341b |
|  | N3 | $y = 1.005/(1 + 16.431e{-}0.192x)$ | 0.997 | 0.048a | 0.028a | 31.235a |
|  | N4 | $y = 0.918/(1 + 18.266e{-}0.195x)$ | 0.992 | 0.045a | 0.025b | 30.719b |
|  | N5 | $y = 0.883/(1 + 19.249e{-}0.200x)$ | 0.991 | 0.044a | 0.024b | 30.032b |

**Notes.**
*, ** significant at the 0.05 and 0.01 probability levels, respectively, ns, no significant. Each data point represents the average of the measured data.
Vmax, the maximum accumulation rate; Vmean, the average accumulation rate; D, the active accumulation period.

respectively. The two growing seasons had similar results, suggesting that the N3 treatment simultaneously improved the grain and protein yield in wheat.

## Correlation analysis of grain yield and protein yield with various factors

The correlations between grain yield, grain protein yield, and other output variables were analyzed (Table 4). The grain and protein yields were found to be significantly and positively correlated with $NAAA_{leaf}$, $NAAA_{stem}$, $NAAA_{spike}$, $NTA_{leaf}$, $TE_{leaf}$, $NTA_{stem}$, $TE_{stem}$, $NTA_{spike}$ and $TE_{spike}$, as well as with nitrogen metabolic enzyme activity levels and grain nitrogen filling parameters. Collectively, these findings indicated that the increased whole-grain and protein yields achieved by the N3 nitrogen application management were due to improved nitrogen transport activity in the plants and nitrogen accumulation in the grains.
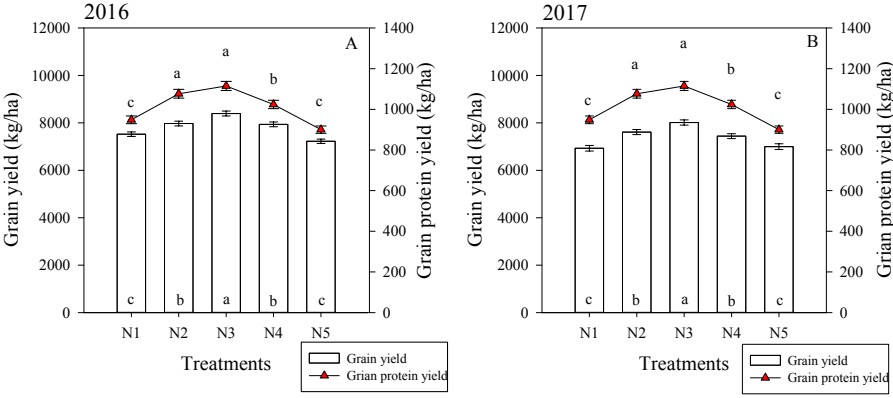

**Figure 5** (A–B) Effects of different treatments on grain yield and grain protein yield at maturity. Each data point represents the average of the measured data.

## DISCUSSION

Wheat plant nitrogen remobilization and partitioning are enhanced at anthesis, translating into proportionally more nitrogen in the ear and less in the vegetative parts of the plant (*Annunziata et al., 2017*; *Noguero & Lacombe, 2016*). In the present study, a decrease in nitrogen accumulation in the vegetative plant parts was detected between anthesis and maturity. This suggested that a major portion of the nitrogen that accumulates in the wheat grains at harvest originates from the degradation of proteins in senescing and falling vegetative segments (*Bouchet et al., 2016*). The application of fertilizer in appropriate proportions can results in greater nitrogen accumulation in the vegetative parts until anthesis (*Liang et al., 2020*). Similarly, we report that the N3 treatment significantly increased the nitrogen accumulation of wheat plants during anthesis. Hence, base or topdressing nitrogen fertilizer application increased less nitrogen in the vegetative organ, while excessive nitrogen fertilizer application decreased nitrogen accumulation during anthesis. The reason for these different effects may be associated with the excessive application of nitrogen fertilizer at one time, aggravating nitrogen lost through leaching or denitrification in the soil. Accumulated nitrogen can be translocated to the grains during grain-filling (*Liu et al., 2020*). In our study, we found that the amount of nitrogen translocated and its contribution to the grain nitrogen were the highest in leaves, followed by stems + sheaths, and then the spike axes + grain husks. This suggested that grain nitrogen was supplied to a greater extent by remobilized nitrogen from leaves and stems. In general, appropriate split nitrogen management (e.g., the N3 treatment in the present study) can significantly promote the transport of nitrogen from the vegetative organs to the grains after anthesis and is conducive to the highest accumulation of nitrogen in grains.

NR and GS are the principal enzymes in plant nitrogen metabolism. Among these, NR is the most rate-limiting enzyme for nitrate assimilation (*Fatholahi, Ehsanzadeh & Karimmojeni, 2020*) because its physiological function is to catalyze the NAD(P)H reduction of nitrate to nitrite. Recent studies indicated that NR is an important point for the integrated control of nitrogen and carbon metabolism (*Ali et al., 2019*). In the
**Table 4  Coefficients of correlation for grain yield and grain protein yield with different.** The bolded text, bold-italicized text and italicized text indicate positive or negative correlations between parameters, respectively. **, and * indicate significance at $P < 0.01$ and $P < 0.05$, respectively.

| R | Grain yield | Grain protein yield |
|---|---|---|
| $NAAA_{leaf}$ | **0.857**$^{**}$ | **0.910**$^{**}$ |
| $NAAM_{leaf}$ | *0.764*$^{*}$ | *−0.702*$^{*}$ |
| $NAAA_{stem}$ | **0.954**$^{**}$ | **0.933**$^{**}$ |
| $NAAM_{stem}$ | −0.534 | −0.602 |
| $NAAA_{spike}$ | ***0.708***$^{*}$ | ***0.710***$^{*}$ |
| $NAAM_{spike}$ | −0.675 | −0.605 |
| $NTA_{leaf}$ | **0.909**$^{**}$ | **0.951**$^{**}$ |
| $TE_{leaf}$ | **0.986**$^{**}$ | **0.989**$^{**}$ |
| $CP_{leaf}$ | 0.266 | 0.369 |
| $NTA_{stem}$ | **0.990**$^{**}$ | **0.987**$^{**}$ |
| $TE_{stem}$ | **0.972**$^{**}$ | **0.986**$^{**}$ |
| $CP_{stem}$ | 0.694 | 0.638 |
| $NTA_{spike}$ | **0.847**$^{**}$ | **0.793**$^{**}$ |
| $TE_{spike}$ | ***0.794***$^{*}$ | ***0.931***$^{*}$ |
| $CP_{spike}$ | 0.698 | 0.627 |
| NAFA | −0.043 | −0.152 |
| CP | *−0.672*$^{*}$ | *−0.750*$^{*}$ |
| NR | **0.896**$^{**}$ | **0.941**$^{**}$ |
| GS | **0.978**$^{**}$ | **0.984**$^{**}$ |
| $FAA_{leaf}$ | **0.989**$^{**}$ | **0.990**$^{**}$ |
| $FAA_{grain}$ | **0.983**$^{**}$ | **0.971**$^{**}$ |
| Vmax | **0.879**$^{**}$ | **0.842**$^{**}$ |
| Vmean | **0.831**$^{**}$ | **0.777**$^{**}$ |
| D | ***0.721***$^{*}$ | ***0.652***$^{*}$ |

**Notes.**

$NAAA_{leaf}$, nitrogen accumulation amount in leaf at anthesis; $NAAM_{leaf}$, nitrogen accumulation amount in leaf at maturity; $NAAA_{stem}$, nitrogen accumulation amount in stem and sheath at anthesis; $NAAM_{stem}$, nitrogen accumulation amount in stem and sheath at maturity; $NAAA_{spike}$, nitrogen accumulation amount in spike axis and grain husk at anthesis; $NAAM_{spike}$, nitrogen accumulation amount in spike axis and grain husk at maturity; $NTA_{leaf}$, nitrogen translocation amount from leaf; $TE_{leaf}$, translocation efficiency from leaf; $CP_{leaf}$, contribution proportion from leaf; $NTA_{stem}$, nitrogen translocation amount from stem and sheath; $TE_{stem}$, translocation efficiency from stem and sheath; $CP_{stem}$, contribution proportion from stem and sheath; $NTA_{spike}$, nitrogen translocation amount from spike axis and grain husk; $TE_{spike}$, translocation efficiency from spike axis and grain husk; $CP_{spike}$, contribution proportion from spike axis and grain husk; $FAA_{leaf}$, free amino acid in flag leaf; $FAA_{grain}$, free amino acid in grain..

*, ** significant at the 0.05 and 0.01 probability levels, respectively, ns, no significant.

present study, treatment with a 50%:50% basal-topdressing nitrogen ratio (i.e., the N3 treatment) resulted in higher NR activity levels in the middle and late stages of grain filling when compared to the other split nitrogen treatments. This may have occurred because an appropriate ratio of basal-topdressing of nitrogen fertilizer can maintain a high soil nitrogen supply capacity in the middle and late stages of wheat filling, as previously reported by *Hu et al. (2016)*. The GS plays an important role in plant nitrogen assimilation, but the physiological process is very complex and varies depending on the growth season length and metabolism (*Hildebrandt et al., 2015*). Improved NR activity reportedly favors GS

activity (*Ali et al., 2019*), which is consistent with the similar results we observed between NR and GS enzyme activity in the present study. Furthermore, the N3 treatment improved the activity of NR and GS, which indicated the regulatory action of split nitrogen fertilizer applications on the nitrogen metabolism in wheat flag leaves. However, excessive base or topdressing nitrogen fertilizer application decreases the activity of NR and GS, potentially because excessive base or topdressing nitrogen fertilizer application increases competition within the wheat population and tillers with ears may accumulate relatively low amounts of nitrogen assimilate, decreasing nitrogen metabolism enzymes activity. Meanwhile, the levels of free amino acids have been shown to vary over time as a result of their use in protein synthesis, degradation, or changes in the activities of nitrogen metabolism enzymes, such as NR or GS (*Hildebrandt et al., 2015*). Similar results were observed in the present study, where treatment with a 50%:50% nitrogen basal-topdressing ratio increased the observed accumulation of free amino acids in these organs. Our experimental results corroborate those of previous studies on wheat and showed that nitrogen application promotes nitrogen metabolism under normal conditions. However, all the treatments we tested in wheat indicated suppressed activities of NR and GS when excessed from nitrogen supplementation was used. Appropriate nitrogen fertilizer application (i.e., the N3 treatment) improved the nitrogen metabolism in flag leaves, which increased the nitrogen uptake efficiency and translocation to the grain, resulting in a well-managed change in the transportation rate of amino acids into protein production (*Hu et al., 2016*).

Wheat grain weight is determined by grain-filling rate and filling time. Grain filling is a key physiological process that determines wheat grain quality (*Zhou et al., 2018*). In comparison to the information on grain-filling only, there is limited knowledge on the grain nitrogen filling characteristics in response to split nitrogen management. In the present study, we measured the accumulation of the 1000-grain nitrogen after anthesis, and fitted the grain nitrogen filling parameters to the Logistic growth equation. The N3 treatment was found to not only increase the average nitrogen accumulation rate but also prolong the active accumulation period relative to the other split nitrogen treatments, which may explain why the highest accumulation of 1000-grain nitrogen was measured under the N3 treatment. These benefits may have resulted from improved root activity, retention of leaf greenness and greater nitrogen accumulation after heading (*Wei et al., 2016*). These results further indicate that the inhibitory effect of excessive base or topdressing nitrogen fertilizer application on grain nitrogen weight is caused mainly by a reduction in the active accumulation period. Previous studies have reported that the active seed filling duration determines the final grain yield. When we compare N3 with other split nitrogen fertilizer treatments, the N3 treatment promoted a higher availability of source nitrogen during grain filling and enhanced the high grain protein yield. In addition, the split nitrogen had no notable effects on the maximum accumulation rates. Collectively, these results suggest that nitrogen applications can regulate the accumulation of 1000-grain nitrogen by affecting the average nitrogen accumulation rate and the active accumulation period.

In the present study, our results indicated that grain yield reached the highest levels in the N3 treatment, intermediate levels in the N2 and N4 treatments, and the lowest levels in the N1 and N5 treatments. These finding are consistent with those of most previous reports

(*Wei et al., 2016*). It is noteworthy that excessive basal and topdressing ratios resulted in lower grain yield. This decrease in yield has two main explanation: (1) excessive basal nitrogen fertilizer causes early leaf senescence due to an insufficient supply of nitrogen in the late growth season (*Luo et al., 2018*; *Gang et al., 2019*), and (2) excessive topdressing nitrogen fertilizer disrupts the balance between vegetative and reproductive growth in wheat, leading to overly strong nutrient uptake and postponed maturity (*Wang et al., 2018*; *Yin et al., 2019*). Our results suggested that split nitrogen fertilizer improved the wheat grain yield through a general increase in the production of protein, thus the observed increase in the protein yield observed in the N3-treated plants. This finding is contradictory to that of *Zheng et al. (2020)*, who described a decrease in grain protein content due to dilution effects. One possible reason for this difference is that the N3 treatment could postpone nitrogen application, which ensured which provides the soil nitrogen distribution in the upper soil profile and increased nitrogen absorption during grain-filling. Similarly, *Li et al. (2019)* found that nitrogen application at a later growing stage in spring wheat guaranteed the leaf function and increased the grain nitrogen accumulation. However, the lower leaf nitrogen metabolizing enzyme activity observed in N4 and N5 treated plants during the later grain-filling could have resulted in a decrease in nitrogen transport from leaf to grain, indicating that higher nitrogen application can decrease grain nitrogen weight and yield. Our study showed that grain nitrogen weight and yield were both significantly higher with the N3 treatment than under the other split nitrogen treatments; therefore, this treatment protocol can be used to increase winter wheat yields. Despite the higher labor costs of split fertilizer application, it is still widely used and believed to be one of the most effective measures for increasing crop yield in the main wheat producing areas worldwide, such as in China. Further studies are required to clarify the mechanism that governs high yield production and high resource use efficiency. In addition, root absorption and soil nitrogen supply capacity should be further investigated.

The present study indicated that post-anthesis nitrogen uptake was low, i.e., only 13.3 kg/ha, which indicated that grain nitrogen was supplied almost exclusively by the remobilized nitrogen from the vegetative organs. Furthermore, the accumulation of nitrogen in the grains was related to the translocation after anthesis (*Nehe et al., 2018b*). We found that grain yield and grain protein yield were significantly positively correlated with $NAAA_{leaf}$, $NAAA_{stem}$, $NAAA_{spike}$, $NTA_{leaf}$, $TE_{leaf}$, $NTA_{stem}$, $TE_{stem}$, $NTA_{spike}$ and $TE_{spike}$. Nitrogen accumulation in vegetative organs at anthesis is likely to be the main physiological driver of nitrogen transport from these organs to developing grains. Our results indicated that grain yield and protein yield were significantly positively correlated with the nitrogen metabolic enzyme activity levels, parameters of grain nitrogen filling. The increases in grain yield and protein yield were attributed primarily to the higher levels of nitrogen transport activity in the plants and grain nitrogen filling, alongside an increase in relevant metabolic activity levels. These results support the use of optimal basal-topdressing fertilizer ratio to achieve high yield and high use efficiency.

## CONCLUSION

In this paper, we have shown that different split nitrogen fertilizer applications can affected grain and protein yields over the two wheat growing seasons. Our study successfully shows that (1) nitrogen fertilization method using a basal-topdressing ratio of 50%:50% is optimal to other ratios and can significantly increase the grain and protein yield; and (2) that these increases could be attributed to the significantly higher nitrogen accumulation levels at anthesis, which improved the nitrogen metabolic enzyme activity levels and nitrogen translocation after anthesis that further increased the nitrogen accumulation in grains during grain filling. We also found that the N3 split nitrogen fertilizer application could affect grain nitrogen filling, thus affecting grain nitrogen accumulation. However, it remains unclear if this was due to improved root absorption or increased soil nitrogen supply capacity. Further research on the process of split nitrogen fertilizer application is needed, which could also provide a better understanding of split nitrogen fertilizer application and its effects on wheat yield and quality.

### Funding

This work was supported by the National Natural Science Foundation of China (31771717), the Natural Science Foundation of Shandong Province (ZR2016CM34), and the Technology System in Modern Wheat Industry, Ministry of Agriculture, China (CARS-3-1-19). The funders had no role in study design, data collection and analysis, decision to publish, or preparation of the manuscript.

### Grant Disclosures

The following grant information was disclosed by the authors:
National Natural Science Foundation of China: 31771717.
Natural Science Foundation of Shandong Province: ZR2016CM34.
Technology System in Modern Wheat Industry, Ministry of Agriculture, China: CARS-3-1-19.

### Competing Interests

The authors declare there are no competing interests.

### Author Contributions

- Zhen Zhang conceived and designed the experiments, performed the experiments, analyzed the data, prepared figures and/or tables, authored or reviewed drafts of the paper, and approved the final draft.
- Zhenwen Yu analyzed the data, authored or reviewed drafts of the paper, funding, and approved the final draft.
- Yongli Zhang conceived and designed the experiments, authored or reviewed drafts of the paper, funding, and approved the final draft.
- Yu Shi analyzed the data, authored or reviewed drafts of the paper, and approved the final draft.

## Data Availability

The raw measurements used to create Figs. 1–5 are available in the Supplementary File.

## Supplemental Information

Supplemental information for this article can be found online at http://dx.doi.org/10.7717/peerj.11467#supplemental-information.

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
