# Peer review of "Optimized nitrogen fertilizer application strategies under supplementary irrigation improved winter wheat (Triticum aestivum L.) yield and grain protein yield"

_PeerJ, doi:10.7717/peerj.11467_

## Round 0.1 · original submission · Major Revisions

Dear Senior editors

The manuscript if revised very carefully could be accepted for publication. However, the manuscript needs major revision as proposed by the three reviewers. The authors will revise the manuscript according to all comments and suggestions and will show their response to each comment of each reviewer.

Reviewer 1 ·

Basic reporting

English should improve by native person. Literature is ok. Hypothesis is OK.

Experimental design

Original primary research is within Aims and Scope of the journal.Research question is well defined.Methods are described with sufficient detail & information to replicate.

Validity of the findings

All underlying data have been provided; they are robust, statistically sound, & controlled.

Conclusions are well stated, linked to original research question & limited to supporting results.

Additional comments

English should improve by a native person. The paper suffers from a poor English structure throughout and cannot be published or reviewed properly in the current format. The manuscript requires a thorough proofread by a native person whose first language is English. The instances of the problem are numerous and this reviewer cannot individually mention them. It is the responsibility of the author(s) to present their work in an acceptable format. Unless the paper is in a reasonable format, it should not have been submitted.
2. The novelty of the study needs to be highlighted compare to other similar studies.
3. Discussion is weak. The discussion needs enhancement with real explanations not only agreements and disagreements. Authors should improve it by the demonstration of biochemical/physiological causes of obtained results. Instead of just justifying results, results should be interpreted, explained to appropriately elaborate inferences. Discussion seems to be poor, didn't give good explanations of the results obtained. I think that it must be really improved. Where possible please discuss potential mechanisms behind your observations. You should also expand the links with prior publications in the area, but try to be careful to not over-reach. For the latter, you should highlight potential areas of future study.
4. The scientific background of the topic is poor. In "Introduction" and "Discussion", the authors should cite recent references between 2016-2020 from JCR journals (with impact factor) about recent achievements.
5. A detailed "Conclusion" should be provided to state the final result that the authors have reached. Please note you only need to place your conclusion and not keep putting results, because these have already been presented in the manuscript.

Reviewer 2 ·

Basic reporting

Clear and unambiguous, professional English used throughout.
Literature references, sufficient field background provided.

Experimental design

This experiment is designed well, the data statistical analysis is ample.

Validity of the findings

All underlying data have been provided, they are robust, statistically sound.
The results are of great interest to the relative readers such as agronomy researchers, field management farmers.

Additional comments

The present MS reported that the results of field experiment conducted with five treatments including different nitrogen fertilizer rates under supplementary irrigation mode to examine the effects of split nitrogen application on grain and protein yields by improving nitrogen translocation, metabolic enzyme activity and grain nitrogen accumulation. The results showed that the use of split nitrogen fertilizer application at a ratio of 50%:50% basal-topdressing significantly increased nitrogen accumulation post-anthesis and nitrogen translocation to grains, increased flag leaves free amino acid levels, activity of nitrate reductase and glutamine synthetase, accumulation rate, active accumulation period, accumulation of 1000-grain nitrogen. The results of this research highlighted the important of this nitrogen application mode on improving grain yield and nitrogen yield together. The present reports add new information relative to their previous reports. Generally, this experiment is designed well, the data statistical analysis is ample, and its results are of great interest to the relative readers such as agronomy researchers, field management farmers. Thus, the paper is of worth to be published in the journal. However, a minor revision may be needed before it can be accepted.

1. I support to change the title to “Optimized nitrogen fertilizer application strategies under supplementary irrigation improved winter wheat (Triticum aestivum L.) yield and grain protein yield”.
2. Please introduce this variety, such as type, planting area, growth duration, general yield, and so on.
3. For the fertilizer application period, it will be better to make it more accurate, such as days after sowing.
4. Please, add more information about the climate for this region.
5. Line 183, please add the detail of this reference “zhang et al.,”
6. “2.3.2. N-metabolizing enzymes”, the measurements of NR and GS should be in different paragraphs.

Figure
Figure 1: I support “percipitation 2016-2017” should be changed to “2016-2017”, “precipitation 2017-2018” should be changed to “2017-2018”.
Table4: the legend of the color is confusing, meanwhile, the values on blue colors are negative, please check again.

Reviewer 3 ·

Basic reporting

no comment

Experimental design

no comment

Validity of the findings

no comment

Additional comments

A two-year field experiment was carried out to investigate optimization of the basal-top-dressing nitrogen ratio for improving winter wheat grain yield and protein yield. The topic is interesting and the findings can provide some new information for the assessment of winter wheat production. At the present pattern, however, there are some minor limitations need to be deal with before publishing.
Line 55-77, it is better to put these two paragraphs together in one paragraph.
Line 94, “covered an of was” should change into “covered area was”.
Line134, these formulas need add to the reference.
Line 183, “zhang et al” should add the published year.
Line 187, “DAA” should be “Days after anthesis (DAA)”, and then used the abbreviation.
Lines 245, "and there no" should be revised into “and there was no”.
Lines 257, deleted “an”.
Line 300, The authors used many abbreviations, which reduced the readability of the manuscript. I would recommend the abbreviations already accepted by the research community.
Figure 1, “Preciptation” should be “Precipitation”.
Figure 4, “Kg/ha” should be “kg/ha”.
Table 3, “R2” should be revised
Check the grammars throughout the whole text to make all the sentences clearer.

---

## Round 0.2 · accepted · Accept

The authors revised the manuscript according to each comment of reviewers. The manuscript looks good and can be accepted for publication in PeerJ, thanks.